# Versatile Tunable Voltage-Mode Biquadratic Filter and Its Application in Quadrature Oscillator

**DOI:** 10.3390/s19102349

**Published:** 2019-05-21

**Authors:** San-Fu Wang, Hua-Pin Chen, Yitsen Ku, Yi-Chun Lin

**Affiliations:** 1Department of Electronic Engineering, National Chin-Yi University of Technology, Taichung 41170, Taiwan; sf_wang@ncut.edu.tw; 2Department of Electronic Engineering, Ming Chi University of Technology, New Taipei 24301, Taiwan; U04157115@mail2.mcut.edu.tw; 3Department of Electrical Engineering, California State University Fullerton, Fullerton, CA 92831, USA.; joshuaku@fullerton.edu

**Keywords:** voltage-mode, analog filter, quadrature oscillator, OTA, circuit theory

## Abstract

This paper presents a versatile tunable voltage-mode biquadratic filter with five inputs and three outputs. The proposed filter enjoys five single-ended output operational transconductance amplifiers (OTAs) and two grounded capacitors. The filter can be easily transformed into a quadrature oscillator. The filter with grounded capacitors is resistorless and electronically tunable. Either a voltage-mode five-input single-output biquadratic filter or a voltage-mode single-input three-output biquadratic filter can be operated by appropriate selecting input and output terminals. In the operation of five-input single-output biquadratic filter, the non-inverting lowpass, non-inverting bandpass, inverting bandpass, inverting highpass, non-inverting bandreject, inverting bandreject, and non-inverting allpass filtering responses can be realized by appropriately applying the input voltage signals. In the operation of single-input three-output biquadratic filter, the non-inverting/inverting lowpass, bandpass and bandreject filtering responses can be realized simultaneously. The circuit provides independent adjustment of the resonance angular frequency and quality factor, high-input impedance, and no inverting-type input voltage signals are imposed. The application in quadrature oscillator exhibits independent electronic tuning characteristic of the oscillation condition and the oscillation frequency. The theoretical analysis has been verified through OrCAD PSpice and furthermore by experimental measurements.

## 1. Introduction

Active filters and oscillators are widely used in sensor, communication, instrumentation and control applications [1,2,3,4,5,6,7,8]. Especially in the sensor system, the filter is used to filter out the noise in the system [1]. Oscillators can produce a repetitive electronic signal for sensors [2,3,4,5,6]. One of the key circuits in front-ends of sensors is the quadrature signal generator for modulation and demodulation [7]. The quadrature signal generator will generate a pair of in-phase and quadrature phase signals for phase shift keying modulation in direct conversion systems. Moreover, the quadrature signals can be used to reduce the signal’s bandwidth to simply the sensor’s design [8].

Active filters with high-input impedance are of great interest because it can be easily cascaded to synthesize high-order filters [9]. Multiple-input single-output universal biquadratic filters with different filter responses are realized according to the selected different input signals, and they can realize all the five standard biquadratic filtering response filters. An attractive feature of single-input three-output multifunction biquadratic filters is that lowpass, bandpass, and highpass/bandreject outputs are simultaneously available in various circuit modes. These additional outputs can be used in systems that employ more than one filter function. The filters using operational transconductance amplifiers (OTAs) with multiple-input single-output or single-input three-output, along with capacitors, a class of OTA-C circuits are the attention for much research [10,11,12,13,14,15,16,17,18]. These designs of OTA-C filter circuits require no resistors, hence, they are suitable for integrated circuit (IC) than the other active components. Moreover, OTA-C filter circuit provides an electronic tunability of its transconductance gain, g_m_, and is also beneficial for compensating the process tolerances.

The recently reported electronically tunable OTA-based multiple-input single-output voltage-mode universal biquadratic filters have been proposed [19,20] that can realize all five standard filtering responses with the following four attractive advantages: (i) orthogonal controllability of the parameters resonance angular frequency (ω_o_) and quality factor (Q), (ii) possession high impedance at input terminal, (iii) all grounded passive components, and (iv) no need to employ inverting-type input voltage signals. The circuit [19] also permits resistorless realization along with two grounded capacitors but the parameters ω_o_ and Q cannot be independently controllable. Although the circuit [20] permits independent of the parameters ω_o_ and Q, it cannot achieve resistorless structure. Both the circuits in [19,20] need to use six OTAs as the active devices and cannot simultaneously realize lowpass, bandpass, and bandreject filters from the same configuration.

This study proposes a versatile voltage-mode biquadratic filter with five inputs and three outputs. The proposed circuit employs five single-ended output OTAs and two grounded capacitors. When operating in five-input single-output universal biquadratic filter, the non-inverting lowpass (LP), non-inverting bandpass (BP), inverting bandpass (IBP), inverting highpass (IHP), non-inverting bandreject (BR), inverting bandreject (IBR), and non-inverting allpass (AP) filtering responses can be obtained by the selections of different input voltage signals. When operating in single-input three-output biquadratic filter, either the LP, IBP, BR filtering responses or the inverting lowpass (ILP), BP and BR filtering responses can be simultaneously realized. Therefore, the circuit realizes single-input three-output biquadratic filter or five-input single-output biquadratic filter into single topology. The circuit also can be transformed into a quadrature oscillator. The proposed circuit enjoys the following advantages: (i) the employment of only five single-ended output OTAs, (ii) the employment of only two grounded capacitors, and thus can absorb equivalent shunt capacitive parasitics, (iii) capability to realize IBP/BP, LP/ILP and BR/IBR filtering responses simultaneously with single-input three-output, (iv) independent control of the LP, ILP, BP and IBP gains simultaneously without affecting the parameters of ω_o_ and Q, (v) realization of all five standard filtering functions with five-input single-output by appropriate selecting different five input voltage signals, (vi) possession high-input impedance, (vii) no need to employ inverting-type input voltage signals, (viii) independent electronic adjustment of the parameters ω_o_ and Q, (ix) easy transformed into a quadrature oscillator, and (x) low active and passive sensitivity performances. Table 1 compares the proposed filter with recently reported voltage-mode OTA-based biquadratic filters [19,20]. Unlike the reported voltage-mode OTA-based biquadratic filters in [19,20], the attractive feature of the proposed filter is that it employs only five single-ended OTAs and two grounded capacitors for realizing voltage-mode biquadratic filter and capability to realize IBP/BP, LP/ILP and BR/IBR filtering responses simultaneously with single-input three-output. Moreover, the LP, ILP, BP and IBP gains can be independently controlled simultaneously without affecting the parameters of ω_o_ and Q. The proposed circuit also can be converted into a quadrature oscillator.

## 2. Circuit Descriptions and Realizations

### 2.1. Proposed Electronically Tunable Voltage-Mode Biquadratic Filter

The proposed electronically tunable voltage-mode biquadratic filter is shown in Figure 1. It uses only five OTAs, two grounded capacitors and no resistors which is beneficial from the IC fabrication point of view. The five input signals, V_i1_, V_i2_, V_i3_, V_i4_, and V_i5_, are connected to the high-input impedance input nodes of OTAs, respectively. Therefore, the proposed circuit enjoys the advantage of having high-input impedance. An ideal OTA has infinite input and output impedances. The output current of an OTA is given by I_O_ = g_m_ (V + −V-) where g_m_ is the transconductance gain of the OTA [19,20]. Routing analysis of the circuit depicted in Figure 1 yields the following OTA-C filter three output voltages.

(1)$$V_{o1}=\frac{-s\frac{g_{m1}}{C_{1}}V_{i1}-s\frac{g_{m5}}{C_{1}}{(V}_{i2}-V_{i3})+\frac{g_{m1}g_{m2}g_{m4}}{C_{1}C_{2}g_{m3}}V_{i4}+s\frac{g_{m1}g_{m4}}{C_{1}g_{m3}}V_{i5}}{D(s)}$$

(2)$$V_{o2}=\frac{\frac{g_{m1}g_{m2}}{C_{1}C_{2}}V_{i1}+\frac{g_{m2}g_{m3}g_{m5}}{C_{1}C_{2}g_{m3}}{(V}_{i2}-V_{i3})+(s\frac{g_{m2}}{C_{2}}+\frac{g_{m1}g_{m2}g_{m4}}{C_{1}C_{2}g_{m3}}{)V}_{i4}-\frac{g_{m1}g_{m2}g_{m4}}{C_{1}C_{2}g_{m3}}V_{i5}}{D(s)}$$

(3)$$V_{o3}=\frac{{(s}^{2}+\frac{g_{m1}g_{m2}}{C_{1}C_{2}}{)V}_{i1}-s\frac{g_{m4}g_{m5}}{C_{1}g_{m3}}{(V}_{i2}-V_{i3})+\frac{g_{m1}g_{m2}g_{m4}}{C_{1}C_{2}g_{m3}}V_{i4}-{(s}^{2}+\frac{g_{m1}g_{m2}g_{m4}}{C_{1}C_{2}g_{m3}}{)V}_{i5}}{D(s)}$$

Denominator D(s) is given by
(4)$$D(s)=s^{2}+s\frac{\omega_{o}}{Q}+\omega_{o}^{2}=s^{2}+s\frac{g_{m1}g_{m4}}{C_{1}g_{m3}}+\frac{g_{m1}g_{m2}}{C_{1}C_{2}}$$

The parameters of ω_o_, Q and bandwidth ω_o_/Q given in Equation (4) are calculated as
(5)$$\omega_{o}=\sqrt{\frac{g_{m1}g_{m2}\;}{C_{1}C_{2}}},\;Q=\frac{g_{m3}}{g_{m4}}\sqrt{\frac{C_{1}g_{m2}}{C_{2}g_{m1}}},\;\frac{\omega_{o}}{Q}=\frac{g_{m1}g_{m4}}{C_{1}g_{m3}}$$

Based on Equation (5), the parameter ω_o_ can be tuned through g_m1_ and g_m2_, and the parameter Q can be adjusted independently through g_m3_ and g_m4_ without disturbing ω_o_. After adjusting ω_o_, the parameter Q can be adjusted independently by the ratio of g_m3_ and g_m4_, and a large Q value can be obtained by adjusting this ratio. In other words, the circuit offers orthogonal control of the parameters of ω_o_ and Q. Note that by selecting g_m1_ = g_m2_ = g_m_, the parameters of ω_o_ and Q in Equation (5) becomes
(6)$$\omega_{o}=g_{m}\sqrt{\frac{1\;}{C_{1}C_{2}}},\;Q=\frac{g_{m3}}{g_{m4}}\sqrt{\frac{C_{1}}{C_{2}}}$$

According to Equation (6), the parameter ω_o_ can be adjusted independently by varying g_m_, while the parameter Q can be adjusted independently by varying g_m3_ and g_m4_. This means that the parameters of ω_o_ and Q of the filter can be independently controlled.

#### 2.1.1. Single-Input Three-Output Voltage-Mode Biquadratic Filter

According to Equations (1)–(3), The IBP, LP and BR signals can be simultaneously obtained by setting the input voltage V_i2_ = V_i3_ = V_i4_ = V_i5_ = 0 (grounded) and getting V_i1_ as input signal V_in_. The three voltage transfer functions obtained are

(7)$$\frac{V_{o1}}{V_{\mathrm{in}}}=\frac{-s\frac{g_{m1}}{C_{1}}}{D(s)}$$

(8)$$\frac{V_{o2}}{V_{\mathrm{in}}}=\frac{\frac{g_{m1}g_{m2}}{C_{1}C_{2}}}{D(s)}$$

(9)$$\frac{V_{o3}}{V_{\mathrm{in}}}=\frac{s^{2}+\frac{g_{m1}g_{m2}}{C_{1}C_{2}}}{D(s)}$$

Thus, the circuit realizes an IBP signal at V_o1_, a LP signal at V_o2_, and a BR signal at V_o3_, simultaneously. The IBR, LP, and BR gain constants in Equations (7)–(9) are

(10)$$H_{\mathrm{IBP}}=-\frac{g_{m3}}{g_{m4}},\;H_{\mathrm{LP}}=H_{\mathrm{BR}}=1$$

Similarly, The BP, ILP, and IBR signals can be simultaneously obtained by setting the input voltage V_i1_ = V_i2_ = V_i3_ = V_i4_ = 0 and getting V_i5_ as input signal V_in_. The three voltage transfer functions obtained are
(11)$$\frac{V_{o1}}{V_{\mathrm{in}}}=\frac{\left(\frac{g_{m4}}{g_{m3}})(s\frac{g_{m1}}{C_{1}}\right)}{D(s)}$$
(12)$$\frac{V_{o2}}{V_{\mathrm{in}}}=\frac{-(\frac{g_{m4}}{g_{m3}})(\frac{g_{m1}g_{m2}}{C_{1}C_{2}})}{D(s)}$$
(13)$$\frac{V_{o3}}{V_{\mathrm{in}}}=\frac{-(\frac{g_{m4}}{g_{m3}}{)(s}^{2}+\frac{g_{m1}g_{m2}}{C_{1}C_{2}})}{D(s)}$$

Thus, the circuit realizes a BP signal at V_o1_, an ILP signal at V_o2_, and an IBR signal at V_o3_, simultaneously. The BP, ILP and IBR gain constants in Equations (11)–(13) are
(14)$$H_{\mathrm{BP}}=1,\;H_{\mathrm{ILP}}=H_{\mathrm{IBR}}=-\frac{g_{m4}}{g_{m3}}$$

Moreover, two IBP signals and a LP signal can be simultaneously obtained by setting the input voltage V_i1_ = V_i3_ = V_i4_ = V_i5_ = 0 and getting V_i2_ as input signal V_in_. The three voltage transfer functions obtained are
(15)$$\frac{V_{o1}}{V_{\mathrm{in}}}=\frac{-s\frac{g_{m5}}{C_{1}}}{D(s)}$$
(16)$$\frac{V_{o2}}{V_{\mathrm{in}}}=\frac{\frac{g_{m2}g_{m5}}{C_{1}C_{2}}}{D(s)}$$
(17)$$\frac{V_{o3}}{V_{\mathrm{in}}}=\frac{-s\frac{g_{m4}g_{m5}}{C_{1}g_{m3}}}{D(s)}$$

Thus, the circuit realizes an IBP signal at V_o1_, a LP signal at V_o2_, and an IBP signal at V_o3_, simultaneously. The two IBP gain constants and a LP gain constant in Equations (15)–(17) are
(18)$$H_{\mathrm{IBP}1}=\frac{-g_{m3}g_{m5}}{g_{m1}g_{m4}},\;H_{\mathrm{LP}}=\frac{g_{m5}}{g_{m1}},\;H_{\mathrm{IBP}2}=\frac{-g_{m5}}{g_{m1}}$$

Based on Equation (18), two IBP gain constants and a LP gain constant can be independently controlled by changing g_m5_ without affecting of ω_o_ and Q. This means that the circuit provides independent control of IBP and LP gains simultaneously without affecting the parameters of ω_o_ and Q.

Similarly, two BP signals and an ILP signal can be simultaneously obtained by setting the input voltage V_i1_ = V_i2_ = V_i4_ = V_i5_ = 0 and getting V_i3_ as input signal V_in_. The three voltage transfer functions obtained are
(19)$$\frac{V_{o1}}{V_{\mathrm{in}}}=\frac{s\frac{g_{m5}}{C_{1}}}{D(s)}$$
(20)$$\frac{V_{o2}}{V_{\mathrm{in}}}=\frac{-\frac{g_{m2}g_{m5}}{C_{1}C_{2}}}{D(s)}$$
(21)$$\frac{V_{o3}}{V_{\mathrm{in}}}=\frac{s\frac{g_{m4}g_{m5}}{C_{1}g_{m3}}}{D(s)}$$

Thus, the circuit realizes a BP signal at V_o1_, an ILP signal at V_o2_, and a BP signal at V_o3_, simultaneously. The gain constants in Equations (19)–(21) are
(22)$$H_{\mathrm{BP}1}=\frac{g_{m3}g_{m5}}{g_{m1}g_{m4}},\;H_{\mathrm{LP}}=\frac{-g_{m5}}{g_{m1}},\;H_{\mathrm{BP}2}=\frac{g_{m5}}{g_{m1}}$$

Based on Equation (22), two BP gain constants and an ILP gain constant can be independently controlled by changing g_m5_ without effecting of ω_o_ and Q. Therefore, according to Equations (18) and (22), the circuit provides independent control of the LP, ILP, BP and IBP gains simultaneously without affecting the parameters of ω_o_ and Q. Table 2 summarizes the single-input three-output voltage-mode biquadratic filter of the transfer functions.

#### 2.1.2. Five-Input Single-Output Voltage-Mode Biquadratic Filter

According to Equation (3), we consider five-input single-output in order to realize five standard filtering functions. Assuming that ${\mathrm{kg}}_{m3}=g_{m1}=g_{m4}=g_{m5}$, where k is a scaling factor, the input-output relationship in Equation (3) can be expressed as
(23)$$V_{o3}=\frac{{(s}^{2}+\frac{1}{\tau_{1}\tau_{2}}{)V}_{i1}-\frac{k}{\tau_{1}}{s(V}_{i2}-V_{i3})+\frac{k}{\tau_{1}\tau_{2}}V_{i4}-{k(s}^{2}+\frac{1}{\tau_{1}\tau_{2}}{)V}_{i5}}{s^{2}+\frac{k}{\tau_{1}}s+\frac{1}{\tau_{1}\tau_{2}}}$$
where in Equation (23), $\tau_{1}=\frac{C_{1}}{g_{m1}}$ and $\tau_{2}=\frac{C_{2}}{g_{m2}}$ are the realized time-constants. As indicated by Equation (23), the LP, BP, IBP, IHP, BR, IBR, and AP filtering functions are implemented by appropriately applying the input signal without inverting-type input signal requirements. Table 3 summarizes the five-input single-output voltage-mode biquadratic of the transfer functions according to Equation (23).

### 2.2. Modification of Proposed Filter as Quadrature Oscillator

It is interesting to note that the circuit of Figure 1 can be converted into a voltage-mode quadrature oscillator simply by connecting node V_o3_ to V_i3_ and letting V_i1_ = V_i2_ = V_i4_ = V_i5_ = 0. Figure 2 shows the proposed voltage-mode quadrature oscillator. The characteristic equation of the voltage-mode quadrature oscillator in Figure 2 is obtained as:
(24)$$s^{2}C_{1}C_{2}g_{m3}+{\mathrm{sC}}_{2}g_{m4}{(g}_{m1}-g_{m5})+g_{m1}g_{m2}g_{m3}=0$$

Inspection of Equation (24) shows that the condition of oscillation (CO) and the frequency of oscillation (FO) are given by
(25)$$\mathrm{CO}:\;g_{m1}\leq g_{m5}$$
(26)$$\mathrm{FO}:\;\omega_{o}=\sqrt{\frac{g_{m1}g_{m2}\;}{C_{1}C_{2}}}$$

As indicated by Equations (25) and (26), the CO can be independently controlled by g_m5_ without disturbing the FO, and the FO can be independently controlled by g_m2_ without disturbing the CO. This means that both CO and FO are independent controllable by different transconductance gains.

Under steady state, the relationship of the two voltage outputs of Figure 2 are as
(27)$$V_{o1}=-j\frac{\omega_{o}C_{2}}{g_{m2}}V_{o2}$$

As indicated by Equation (27), the voltages V_o1_ and V_o2_ are in the quadrature form. Thus, the proposed circuit in Figure 1 can be easily transformed into a quadrature oscillator.

## 3. Effect of the OTA Parasitic Elements of the Proposed Voltage-Mode Biquadratic Circuit

The effect of OTA parasitic elements on the performance of voltage-mode filter is considered. Figure 3 depicts the simplified equivalent circuit that represents a non-ideal OTA with finite parasitic resistances and capacitances [20]. R_pj_, R_nj_, and R_oj_ are the input and output terminals parasitic resistances of the jth OTA, respectively. C_pj_, C_nj_, and C_oj_ are the input and output terminals parasitic capacitances of the jth OTA, respectively. In practical OTA’s the input and output parasitic conductance are very smaller the transconductance g_m_. In the presence of these parasitic elements, the circuit presented in Figure 1 is modified to become the circuit presented in Figure 4, in which C_1p_ = C_o1_//C_n2_//C_n4_//C_o5_, C_2p_ = C_p1_//C_o2_, C_3p_ = C_n1_//C_n3_//C_o4_, R_1p_ = R_o1_//R_n2_//R_n4_//R_o5_, R_2p_ = R_p1_//R_o2_, and R_3p_ = R_n1_//R_n3_//R_o4_. It is further noted that the proposed circuit employs external capacitors C_1_ and C_2_ parallel connecting at the output nodes V_o1_ and V_o2_, respectively. As a result, the effects of the parasitic capacitances C_1p_ and C_2p_ can be absorbed, because C_1_ >> C_1p_, and C_2_ >> C_2p_. The parasitic capacitance C_1p_ can be absorbed in the external capacitance C_1_, but the presence of parasitic resistance at V_o1_ output terminal would change the type of the impedance, which should be of a purely capacitive character. The parasitic capacitance C_2p_ can be absorbed in the external capacitance C_2_, but the presence of parasitic resistance at V_o2_ output terminal also would change the type of the impedance. To reduces its effect, one possible solution is to make the operating frequency $\omega_{o}>>\max\{\frac{1}{{(C}_{1}+C_{1p}{)R}_{1p}},\frac{1}{{(C}_{2}+C_{2p}{)R}_{2p}}\}$. The parasitic impedance at V_o3_ output terminal is connected a grounded impedance (R_3p_//C_3p_). This fact affects the operating frequency in the high frequency region, because C_3p_ = (C_n1_ + C_n3_ + C_o4_). To alleviate the effects of parasitic impedance at V_o3_ output terminal, the frequency of operation must be smaller than $\frac{1}{C_{3p}R_{3p}}$. To minimize the effects of the OTAs’ non-idealities, the useful oscillation frequency range of the proposed filter is limited by the following conditions:(28)$$\max\{\frac{1}{{(C}_{1}+C_{1p}{)R}_{1p}},\frac{1}{{(C}_{2}+C_{2p}{)R}_{2p}}\}<<\omega_{o}<<\frac{1}{C_{3p}R_{3p}}$$

It is not difficult to satisfy these conditions, the external capacitances C_1_ and C_2_ can be set much greater than the input and output parasitic capacitances. If the conditions of $\frac{1}{{s(C}_{1}+C_{1p})}<<R_{1p}$, $\frac{1}{{s(C}_{2}+C_{2p})}<<R_{2p}$ and $g_{m3}>>{\mathrm{sC}}_{3p}+\frac{1}{R_{3p}}$ are satisfied, the influence of OTA parasitic elements on the proposed filter in Figure 1 can be ignored.

The active and passive sensitivities of the proposed circuits are low and not larger then unity in absolute value.

## 4. Simulation and Experimental Results

According to LT1228 datasheet [21], the LT1228 consists of two amplifiers to form a current feedback amplifier, which can operate from any supply voltage of ±2 V to ±15 V. The LT1228 implements gain control with a transconductance amplifier and its gain is proportional to an external controlled current. The LT1228 current feedback amplifier is an excellent buffer for transconductance amplifiers due to its very high input impedance. In addition, the LT1228 maintains a wide bandwidth over the wide voltage gain range of the current feedback amplifier that makes the transconductance amplifier output easy to connect to other circuits. The LT1228 is designed to drive low impedance loads like other current feedback amplifiers. The input signal should be less than 100 mVp to match the dynamic range of the transconductance amplifier. The LT1228 consists of transconductance amplifier and current feedback amplifier, where the transconductance amplifier converts the current into a voltage, and then the current feedback amplifier amplifies the voltage. Two specific sensor applications have been proposed in [22,23]. The proposed filter is built by LT1228, so the proposed filter can operate between ±2 V and ±15 V. The low voltage filter such as ±2 V filter can be applied to biomedical sensor systems and internet of things, and the high voltage filter such as ±15 V filter can be applied to automotive electronic sensing devices and industrial electronic sensing devices.

The performance of the proposed OTA-based voltage-mode filter and oscillator has been evaluated using OrCAD PSpice simulations based on macro-model of LT1228 OTA commercially available IC. According to LT1228 datasheet [21], the transconductance gain, g_m_, is given by the formula:(29)$$g_{m}={10\;I}_{\mathrm{SET}}$$
where I_SET_ is the bias current of LT1228. This feature makes it useful for electronic control of transconductance gain. From the internal structure of LT1228 OTA, the current can be written as
(30)$$I_{\mathrm{SET}}=\frac{V_{\mathrm{DD}}-V_{\mathrm{SS}}-{2V}_{\mathrm{BE}}}{R_{\mathrm{tun}}}$$
where V_BE_ in bipolar junction transistor is about 0.65 V and R_tun_ is an external resistor used for adjusting the current I_SET_, while V_DD_ and V_SS_ are the positive and negative bias voltages, respectively. Thus, the value of the resistor R_tun_ will be utilized to implement the desired values of the transconductance gain and make the transconductance gain electronically tunable.

### 4.1. Proposed Voltage-Mode Biquadratic Filter Simulation Results

OrCAD PSpice simulations based on macro-model LT1228 ICs were performed to verify the workability of the OTA-based circuit. The supply voltages were V_DD_ = −V_SS_ = 15 V, and the bias currents 100 µA (i.e., g_m_ = 10 I_SET_ [21,24]) for OTA_1_ to OTA_5_. All transconductances and capacitances were given as 1 mS and 1 nF, respectively, and were designed to obtain a OTA-based circuit in Figure 1 with an angular frequency of f_o_ = 159.16 kHz, and a quality factor of Q = 1. Figure 5 shows the simulated gain and phase responses for the IBP (V_o1_), LP (V_o2_), and BR (V_o3_) filters in Figure 1 with V_i1_ = V_in_, and V_i2_ = V_i3_ = V_i4_ = V_i5_ = 0. Figure 6 shows the simulated gain and phase responses for the BP (V_o1_), ILP (V_o2_), and IBR (V_o3_) filters in Figure 1 with V_i5_ = V_in_, and V_i2_ = V_i3_ = V_i4_ = V_i5_ = 0. Figure 7 shows the five-input single-output voltage-mode biquadratic filter simulated gain responses at the V_o3_ output terminal of Figure 1 by appropriate selecting different five input voltage signals. The values of C_1_ = C_2_ = 1 nF, g_m3_ = g_m4_ = g_m5_ = 1 mS (I_SET_ = 100 uA), Q = 1 were maintained, and simultaneously, only g_m1_ and g_m2_ values were varied with different values of 1 mS, 1.5 mS (I_SET_ = 150 uA), 2 mS (I_SET_ = 200 uA), and 3 mS (I_SET_ = 300 uA); this resulted in BP responses at the V_o1_ output terminal with simulated angular frequencies f_o_ = 156.68 kHz, f_o_ = 235.03 kHz, f_o_ = 313.72 kHz, and f_o_ = 472.03 kHz, with errors of −1.56%, −1.55%, −1.44%, and −1.14%, respectively. The resulting BP filter responses with V_i5_ = V_in_, and V_i1_ = V_i2_ = V_i3_ = V_i4_ = 0 at the V_o1_ output terminal are shown in Figure 8a, which exhibited tuning of f_o_ without affecting Q. Next, the parameter Q was tuned without disturbing f_o_ by keeping the values g_m1_ = g_m2_ = g_m4_ = g_m5_ and varying g_m3_. Figure 8b shows the tunability of Q in the BP responses with V_i5_ = V_in_, and V_i1_ = V_i2_ = V_i3_ = V_i4_ = 0 at the V_o1_ output terminal by varying only transconductance g_m3_ value. As an example, values were selected to realise a constant angular frequency of f_o_ = 159.16 kHz, the component values C_1_ = C_2_ = 1 nF, g_m1_ = g_m2_ = g_m4_ = g_m5_ = 1 mS and only varying the value g_m3_ = 1 mS, g_m3_ = 1.5 mS, g_m3_ = 2 mS and g_m3_ = 3 mS, resulting in Q = 1, 1.5, 2 and 3, respectively. These results demonstrated that tuning of the Q-value without affecting the f_o_-value could be performed using various value of g_m3_. To test the input dynamic range of the filter, the simulation was repeated for a sinusoidal input signal at f_o_ = 159.16 kHz. Figure 9a shows that the input dynamic range of the IBP response with V_i1_ = V_in_, and V_i2_ = V_i3_ = V_i4_ = V_i5_ = 0 at the V_o1_ output terminal with all transconductance values as 1 mS, and C_1_ = C_2_ = 1nF, which extended to an amplitude of 0.1 V (peak) without signification distortion. In Figure 9a, the percentage of the total harmonic distortion (THD) is 2.01%. The dependence of the output harmonic distortion of IBP response on input voltage amplitude is illustrated in Figure 9b. Furthermore, the intermodulation distortion (IMD) for IBP response with V_i1_ = V_in_, and V_i2_ = V_i3_ = V_i4_ = V_i5_ = 0 of filter output voltage V_o1_ in Figure 1 is investigated. Two closely spaced tones, f_1_ = 158 kHz and f_2_ = 160 kHz, were used with equal input signal amplitudes. Figure 10 shows the dependence of the third-order IMD of IBP filter output voltage V_o1_ with two input tones.

### 4.2. The Monte-Carlo Simulations

To collect statistical data regarding mismatch and the variation effect, Monte-Carlo simulations were conducted. The device mismatch was modelled as a set of randomly generated samples that represented the probability distributions of the device parameters. The proposed filter was investigated using Mont-Carlo analysis. Monte-Carlo analysis results for 100 simulations, regarding the IBP frequency responses with V_i1_ = V_in_, and V_i2_ = V_i3_ = V_i4_ = V_i5_ = 0 at the V_o1_ output terminal with all transconductance values given as 1 mS, and C_1_ = C_2_ = 1nF, in which two capacitances C_1_ and C_2_ had a variation of 5% Gaussian deviation. Figure 11 shows the histogram of the angular frequency obtained from the Monte-Carlo analysis.

### 4.3. Modification of Proposed Filter as Quadrature Oscillator Simulations

In order to confirm the theoretical study, the proposed voltage-mode quadrature oscillator realized in Figure 2 was also simulated using OrCAD PSpice simulation program, and the macro-model LT1228 ICs in Figure 2 was used. The supply voltages were V_DD_ = −V_SS_ = 15 V, and the bias currents 100 µA for OTA_1_ to OTA_5_. The quadrature oscillator was designed with g_m1_ = g_m2_ = g_m3_ = g_m4_ = g_m5_ = 1 mS, and C_1_ = C_2_ = 5 nF for the oscillation frequency of f_o_ = 31.83 kHz, in which g_m5_ is larger than the theoretical value to ensure that oscillator will work. The simulation result of quadrature outputs, V_o1_ and V_o2_, is shown in Figure 12a. Figure 12b shows the lissajous pattern of V_o__1_ and V_o__2_ outputs of the simulation results.

### 4.4. Proposed Voltage-Mode Biquadratic Filter Experimental Results

The proposed filter in Figure 1 was experimental tested. The LT1228s commercially available OTAs were used in Figure 1 with the bias currents 100 µA for OTA_1_ to OTA_5_. The component values used in Figure 1 were g_m1_ = g_m2_ = g_m3_ = g_m4_ = g_m5_ = 1 mS, C_1_ = C_2_ = 1 nF, and were designed to obtain a voltage-mode filter with an angular frequency of f_o_ = 159.16 kHz. The subsequent experimental measurements were carried out using keysight E5061B-3L5 network analyzer. Figure 13 shows the measurement gain responses for the IBP (V_o1_), non-inverting LP (V_o2_), and non-inverting BR (V_o3_) filters in Figure 1 with V_i1_ = V_in_, and V_i2_ = V_i3_ = V_i4_ = V_i5_ = 0. Figure 14 shows the measurement LP, IBP, IHP and BR gain responses at the V_o3_ output terminal of Figure 1 by appropriate selecting different five input voltage signals. Figure 15 shows the measurement gain and phase responses for the AP (V_o3_) filter in Figure 1 with V_i1_ = V_i2_ =V_in_, and V_i3_ = V_i4_ = V_i5_ = 0. The measured of electronic tuning the f_o_-value of the BP responses with V_i5_ = V_in_, and V_i1_ = V_i2_ = V_i3_ = V_i4_ = 0 at the V_o1_ output terminal is demonstrated in Figure 16. In Figure 16, the values of C_1_ = C_2_ = 1 nF, g_m3_ = g_m4_ = g_m5_ = 1 mS, Q = 1 were maintained, and simultaneously, only g_m1_ and g_m2_ values were varied with different values of 1 mS, 1.5 mS, 2 mS, and 3 mS; this resulted in BP responses at the V_o1_ output terminal with measured angular frequencies f_o_ = 161.05 kHz, f_o_ = 244.26 kHz, f_o_ = 330.16 kHz, and f_o_ = 505.34 kHz, with errors of 1.19%, 2.32%, 3.72%, and 5.84%, respectively. Figure 17 shows the measured of the Q-tuning without affecting the f_o_-value. In Figure 17, the component values C_1_ = C_2_ = 1 nF, g_m1_ = g_m2_ = g_m4_ = g_m5_ = 1 mS and only varying the value g_m3_ = 1 mS, g_m3_ = 1.5 mS, g_m3_ = 2 mS and g_m3_ = 3 mS, resulting in Q = 1, 1.5, 2 and 3, respectively.

To represent the linearity of proposed filter, the 1-dB power gain compression point (P1dB) is measured through the Agilent N9000A CXA signal analyzer. Figure 18 shows the measured of P1dB of the IBP filter output voltage V_o1_ with V_i1_ = V_in_, and V_i2_ = V_i3_ = V_i4_ = V_i5_ = 0 by applying the input power at the angular frequency of 159 kHz. As shown in Figure 18, the measured P1dB of IBP filter is about −9.2 dBm with respect to output power. Figure 19 and Figure 20 show the spectrum of the IBP filter output voltage V_o1_ through inter-modulation characterization by applying two-tone signals near 159 kHz. In Figure 19, two closely spaced tones, f_1_ = 158 kHz and f_2_ = 160 kHz, were used with equal input amplitudes of 30 mVp. The result shows that the third-order IMD is around −42.35 dBc and the third-order intercept (TOI) is around −5.66 dBm. Figure 20 shows the measured of third-order IMD is around −23.16 dBc and the TOI is around −8.531 dBm with equal input amplitudes of 70 mVp. In Figure 13, Figure 14, Figure 15, Figure 16, Figure 17, Figure 18, Figure 19 and Figure 20, the directly current (DC) power supply is ±15 V

To demonstrate that the proposed circuit can operate on a ±2 V DC supply, Figure 21, Figure 22 and Figure 23 show the results of the operation of the ±2 V DC supply in Figure 13, Figure 14 and Figure 15, which show that the proposed circuit can operate on a ±2 V DC supply. Figure 24 shows the measured of P1dB of the IBP filter output voltage V_o1_ with ±2 V DC power supply by applying the input power at the angular frequency of 159 kHz. As shown in Figure 24, the measured P1dB of IBP filter is about −14.6 dBm with respect to output power. Figure 25 shows the measured of third-order IMD is around −42.86 dBc and the TOI is around −5.474 dBm with equal input amplitudes of 30 mVp when ±2 V DC supply.

## 5. Conclusions

In this paper, a new versatile tunable voltage-mode biquadratic filter with resistorless structure is proposed. The proposed circuit can be used either a single-input three-output biquadratic filter or a five-input single-output biquadratic filter with the same topology, which is more versatile than the filter one with a single input and multiple outputs or the filter one with multiple inputs and single output. The advantages of the proposed circuit are that: (i) the circuit uses only five single-ended output OTAs, two grounded capacitors and no resistors, which is attractive for its IC implementation; (ii) the circuit realizes five generic filter signals without any inverting-type voltage inputs; (iii) the circuit has high-input impedance good for cascadeability the voltage-mode circuits; (iv) the functionality of LP, BP, IBP, IHP, BR, IBR, and AP filtering functions can be easily obtained by appropriate selecting different voltage inputs; (v) the IBP/BP, LP/ILP and BR/IBR filtering responses are simultaneously available without component matching condition; (vi) independent control of the LP, ILP, BP and IBP gains simultaneously without affecting the parameters of ω_o_ and Q, (vii) the parameters ω_o_ and Q can be independently tunable controlled; (viii) the filter topology provides flexible modification of quadrature oscillator; (ix) low active and passive sensitivity performances. OrCAD PSpice simulations and experimental measurements using commercially available LT1228 ICs confirm the feasibility of the proposed filter.

## Figures and Tables

**Figure 1 sensors-19-02349-f001:**
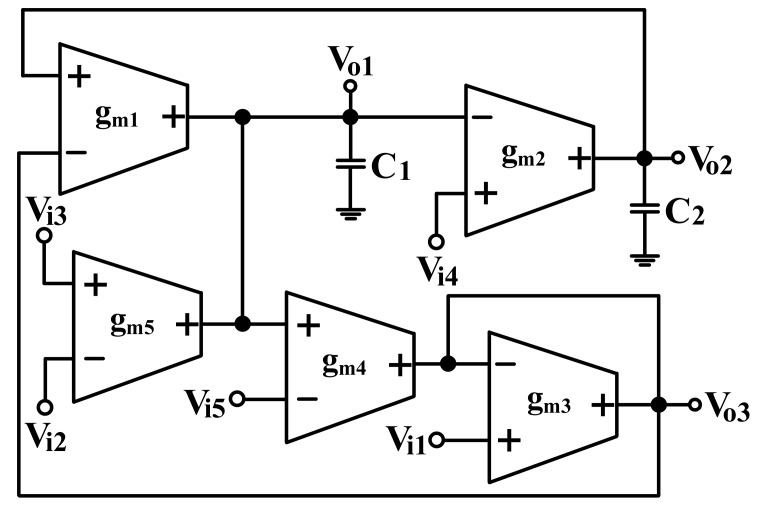
Proposed electronically tunable voltage-mode biquadratic filter.

**Figure 2 sensors-19-02349-f002:**
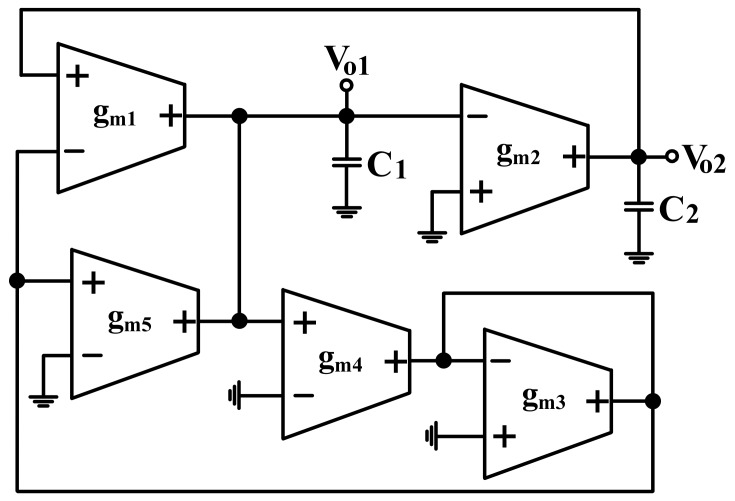
Modified quadrature oscillator of Figure 1.

**Figure 3 sensors-19-02349-f003:**
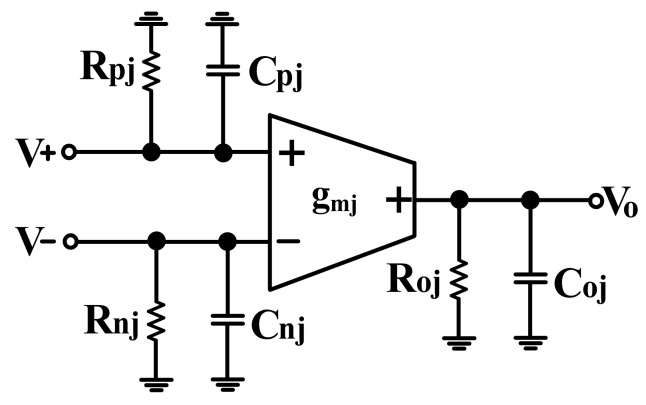
Non-ideal equivalent circuit model of OTA.

**Figure 4 sensors-19-02349-f004:**
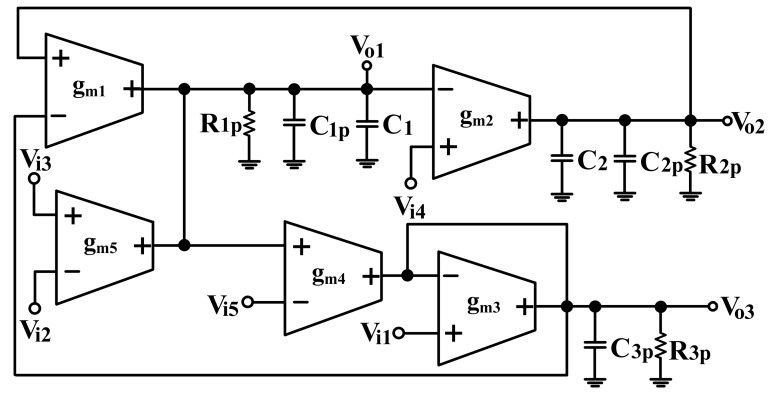
Proposed voltage-mode filter including the parasitic elements of OTA.

**Figure 5 sensors-19-02349-f005:**
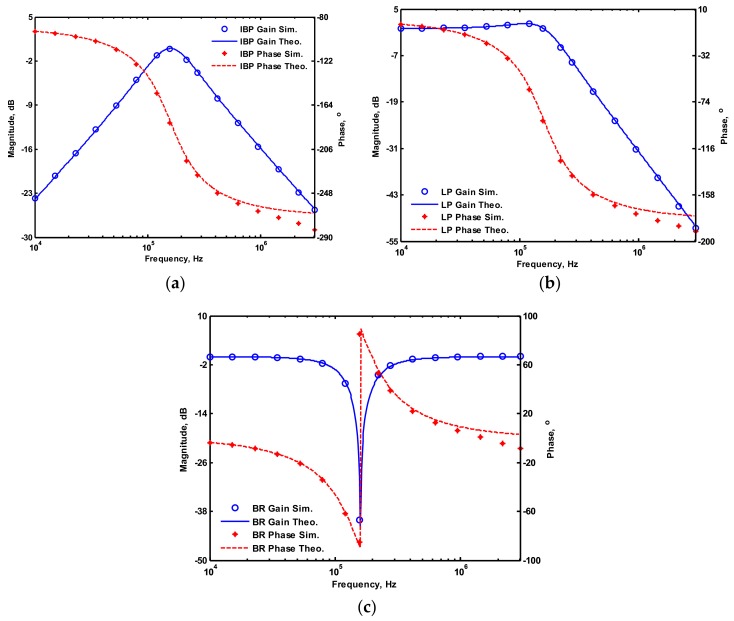
Simulated gain and phase responses of Figure 1 with V_i1_ = V_in_, and V_i2_ = V_i3_ = V_i4_ = V_i5_ = 0 (**a**) inverting bandpass (IBP) filter at V_o__1_ terminal, (**b**) LP filter at V_o__2_ terminal, (**c**) BR filter at V_o__3_ terminal.

**Figure 6 sensors-19-02349-f006:**
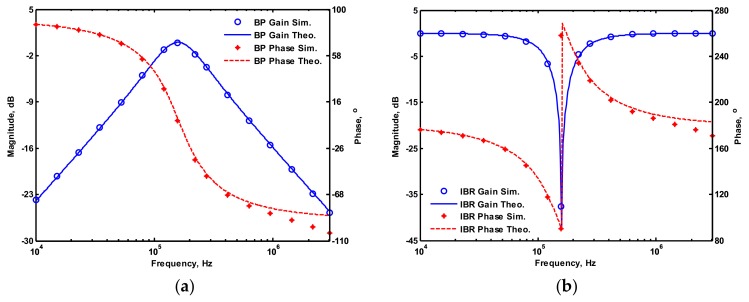
Simulated gain and phase responses of Figure 1 with V_i5_ = V_in_, and V_i1_ = V_i2_ = V_i3_ = V_i4_ = 0 (**a**) BP filter at V_o__1_ terminal, (**b**) inverting lowpass (ILP) filter at V_o__2_ terminal, (**c**) inverting bandreject (IBR) filter at V_o__3_ terminal.

**Figure 7 sensors-19-02349-f007:**
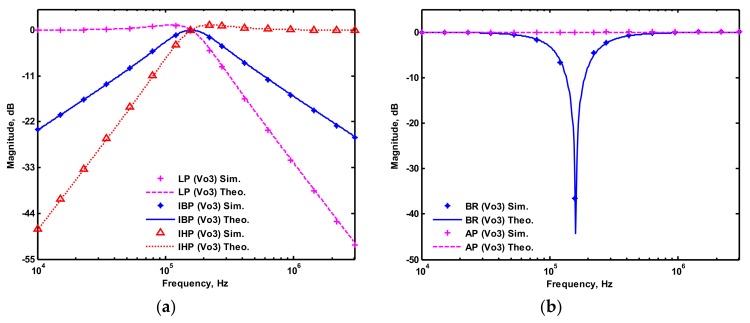
Simulated gain responses of five-input single-output biquad filter at V_o3_ output terminal of Figure 1 (**a**) LP, IBP, and IHP filters, (**b**) BR and AP filters.

**Figure 8 sensors-19-02349-f008:**
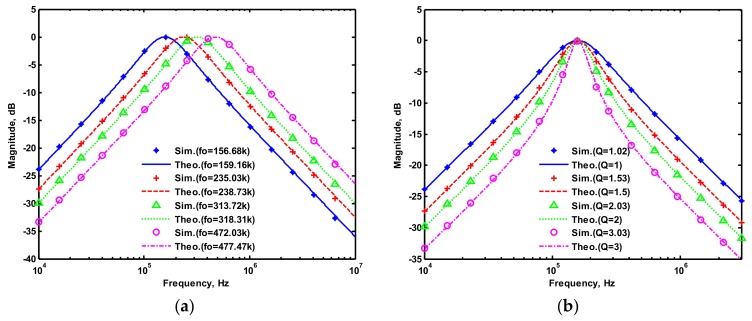
Gain response of the BP (V_o1_) filters, (**a**) varying f_o_ while keeping Q, (**b**) varying Q while keeping f_o_.

**Figure 9 sensors-19-02349-f009:**
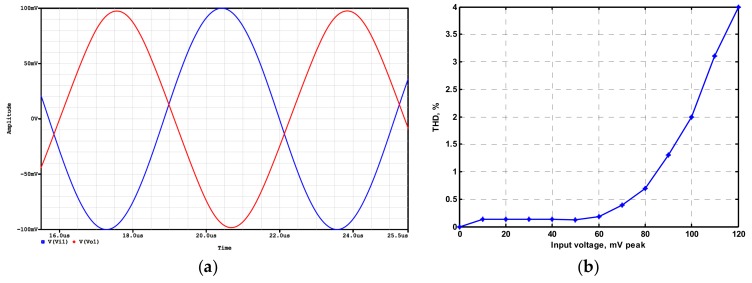
Time-domain results of the IBP (V_o1_) response (**a**) input (blue line) and output (red line) waveforms, (**b**) total harmonic distortion (THD) analysis results.

**Figure 10 sensors-19-02349-f010:**
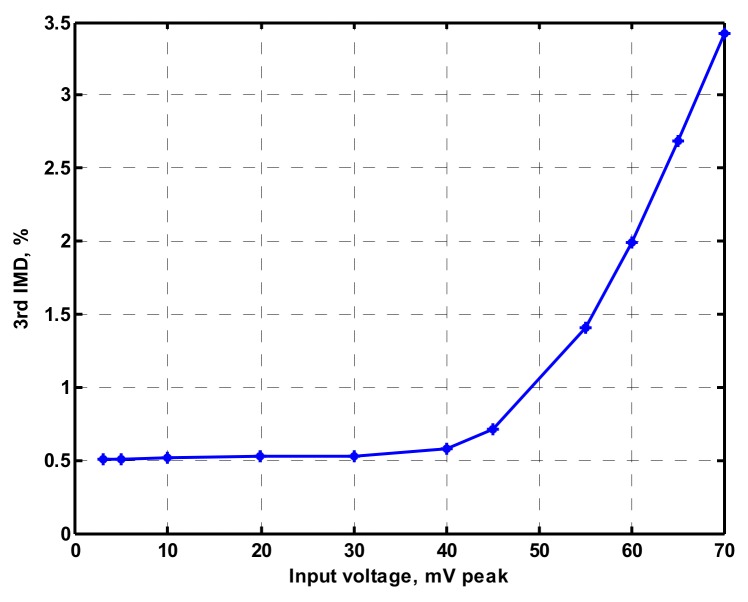
Dependence of the third-order IMD of the IBP filter on input voltage amplitudes.

**Figure 11 sensors-19-02349-f011:**
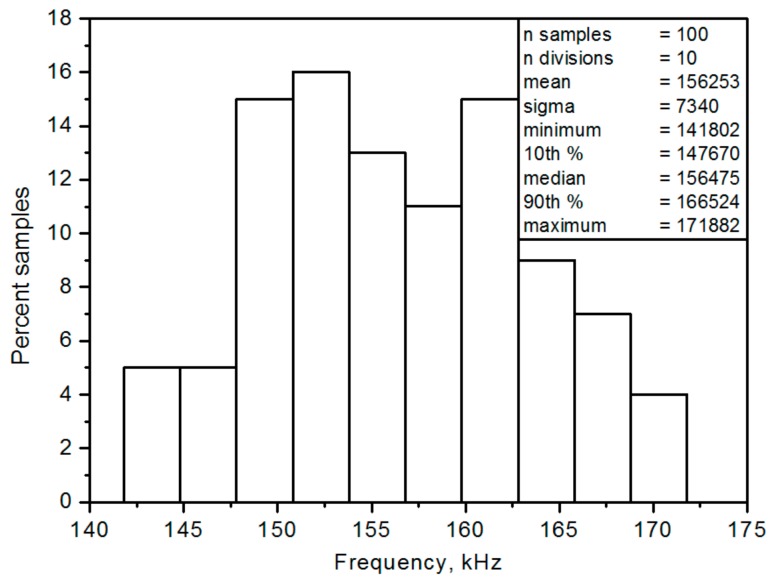
Histogram of Monte-Carlo analysis for IBP filter output voltage V_o1_ terminal.

**Figure 12 sensors-19-02349-f012:**
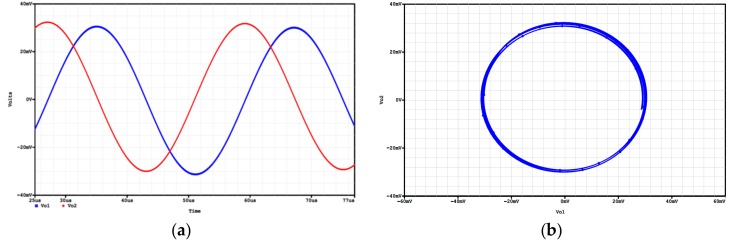
Simulated quadrature voltage outputs in Figure 2 (**a**) V_o1_ (blue) and V_o2_ (red), (**b**) X-Y plot.

**Figure 13 sensors-19-02349-f013:**
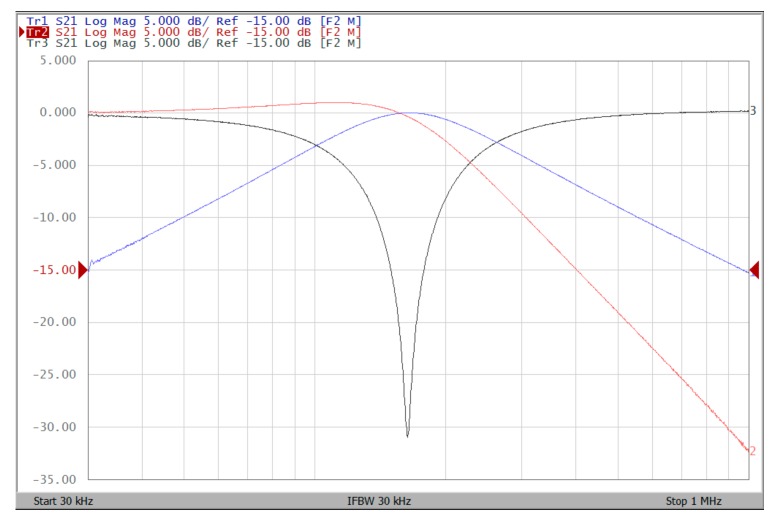
Experimental results of gain responses for IBP (blue), LP (red), and BR (black) filters with V_i1_ = V_in_, and V_i2_ = V_i3_ = V_i4_ = V_i5_ = 0 when ± 15V DC supply.

**Figure 14 sensors-19-02349-f014:**
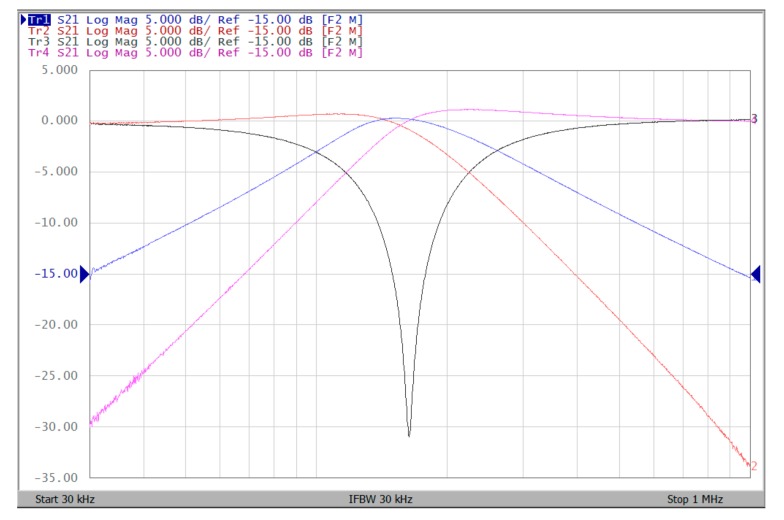
Experimental results of five-input single-output biqaudratic filter at V_o3_ output terminal of Figure 1 by appropriate applied input voltage signals when ±15 V DC supply.

**Figure 15 sensors-19-02349-f015:**
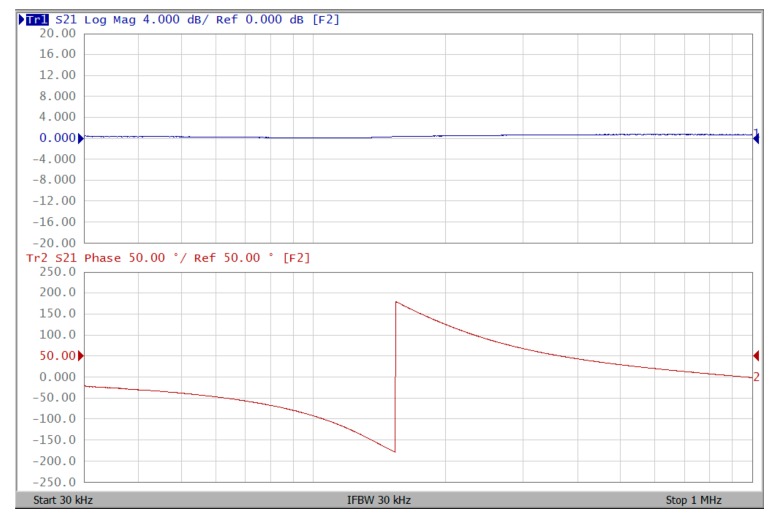
Experimental results of gain (blue) and phase (red) responses for AP filter at V_o3_ with V_i1_ = V_i2_ = V_in_, and V_i3_ = V_i4_ = V_i5_ = 0 when ± 15 V DC supply.

**Figure 16 sensors-19-02349-f016:**
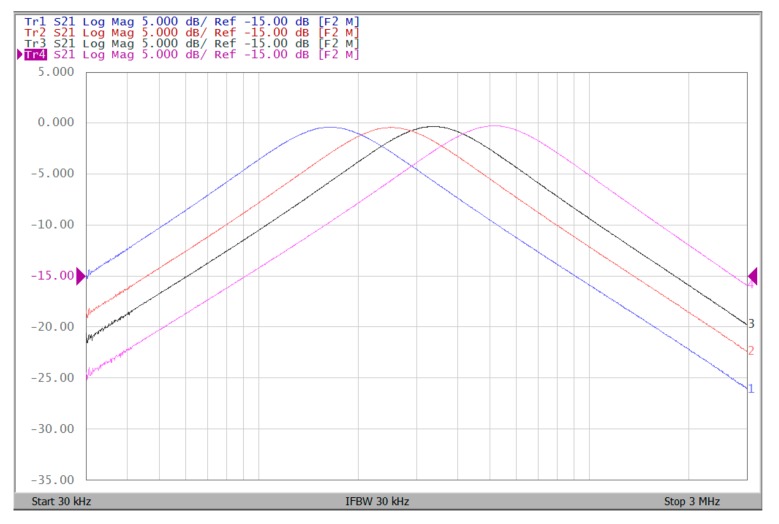
Experimental results of the f_o_-tuning without affecting the Q-value when ±15 V DC supply.

**Figure 17 sensors-19-02349-f017:**
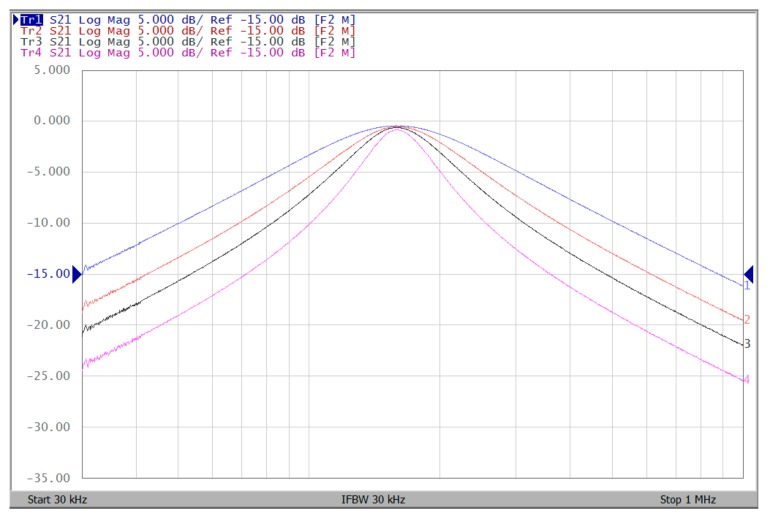
Experimental results of the Q-tuning without affecting the f_o_-value when ±15 V DC supply.

**Figure 18 sensors-19-02349-f018:**
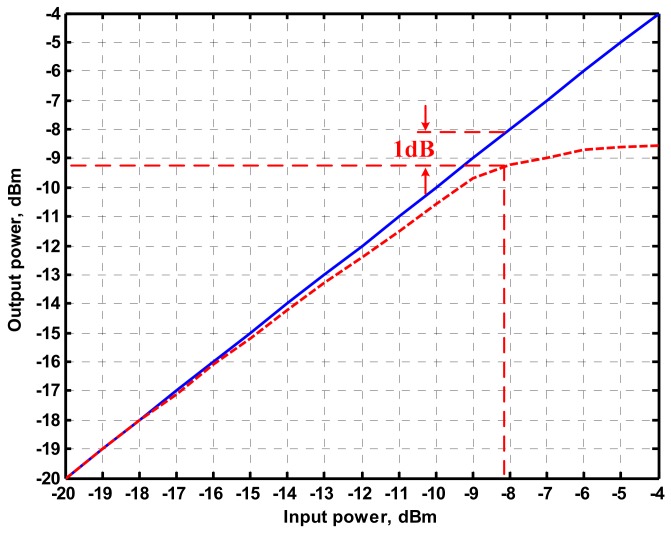
Measured of P1dB of the IBP filter with input power at the V_o1_ when V_i1_ = V_in_, and V_i2_ = V_i3_ = V_i4_ = V_i5_ = 0 when ± 15V DC supply.

**Figure 19 sensors-19-02349-f019:**
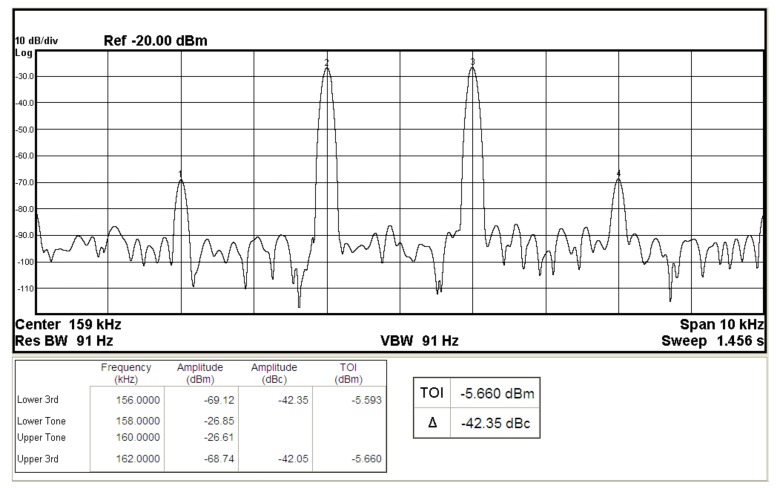
The IBP filter output spectrum for a two-tone intermodulation distortion test with equal input amplitudes of 30 mVp when ±15 V DC supply.

**Figure 20 sensors-19-02349-f020:**
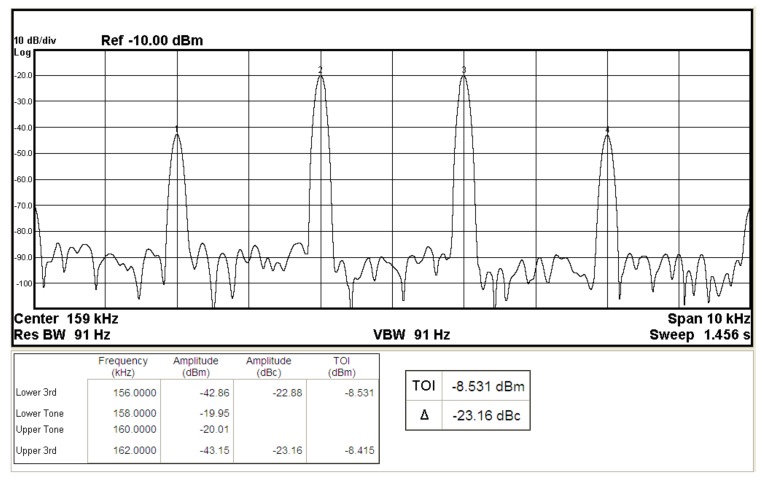
The IBP filter output spectrum for a two-tone intermodulation distortion test with equal input amplitudes of 70 mVp when ±15 V DC supply.

**Figure 21 sensors-19-02349-f021:**
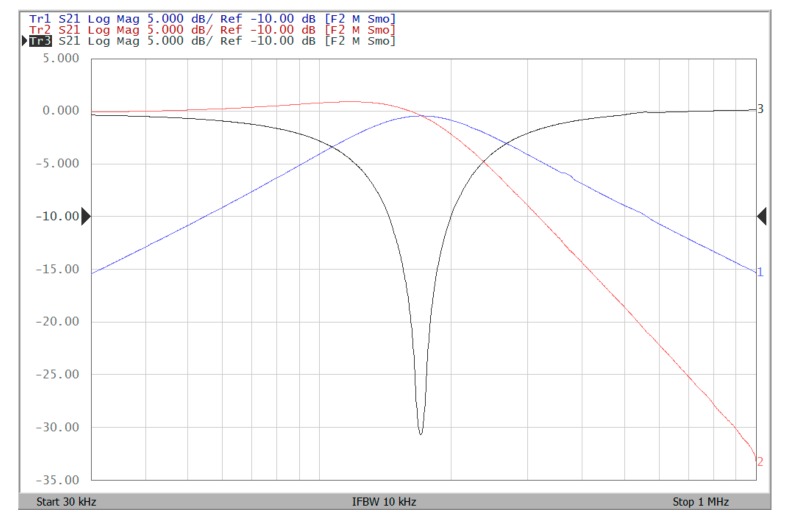
Experimental results of gain responses for IBP (blue), LP (red), and BR (black) filters when ±2 V DC supply.

**Figure 22 sensors-19-02349-f022:**
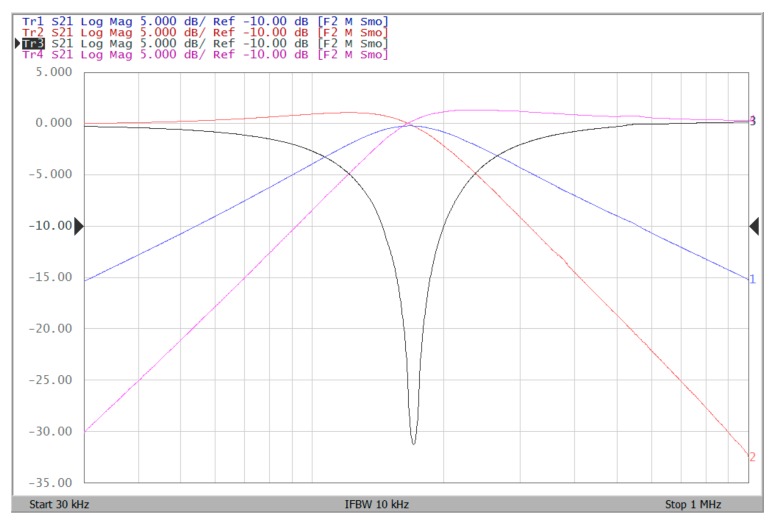
Experimental results of five-input single-output biqaudratic filter at V_o3_ output terminal of Figure 1 by appropriate applied input voltage signals when ±2 V DC supply.

**Figure 23 sensors-19-02349-f023:**
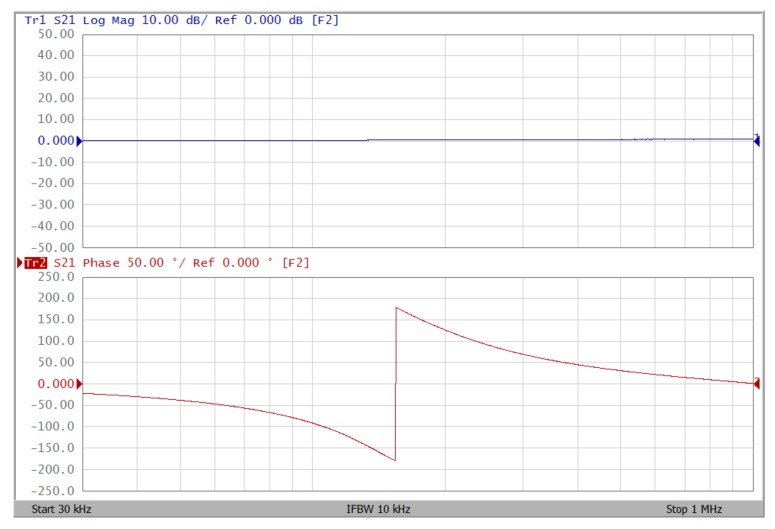
Experimental results of gain (blue) and phase (red) responses for AP filter at V_o3_ when ±2 V DC supply.

**Figure 24 sensors-19-02349-f024:**
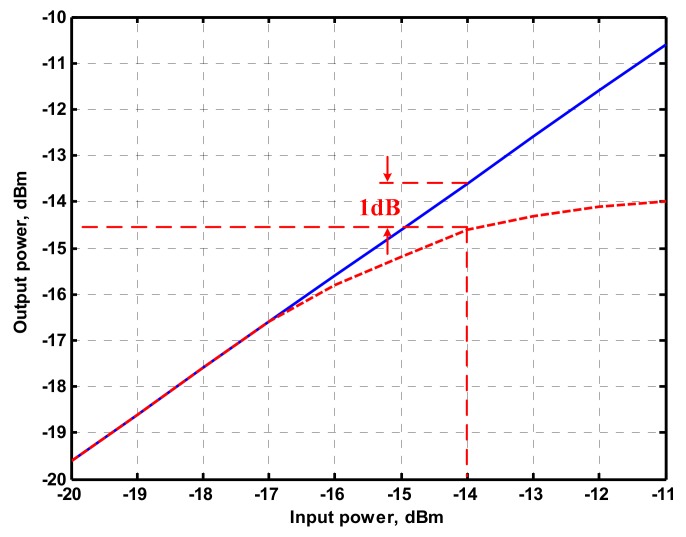
Measured of P1dB of the IBP filter with input power at the V_o1_ when ±2 V DC supply.

**Figure 25 sensors-19-02349-f025:**
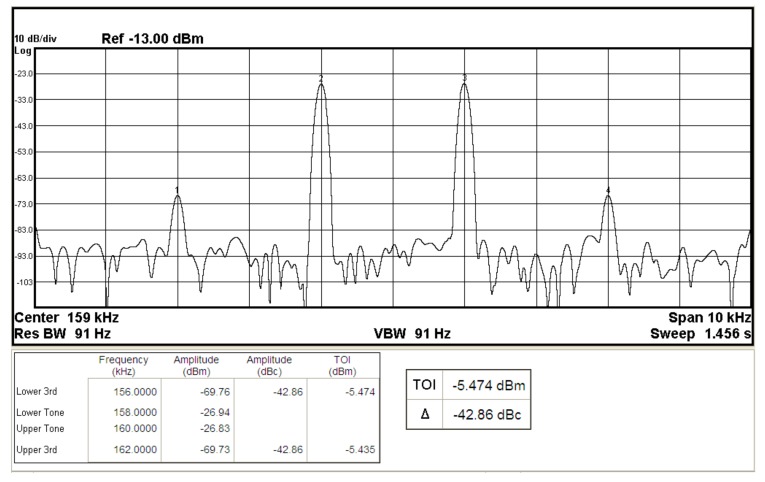
The IBP filter output spectrum for a two-tone intermodulation distortion test with equal input amplitudes of 30 mVp when ±2 V DC supply.

**Table 1 sensors-19-02349-t001:** Comparison of recently proposed operational transconductance amplifier (OTA)-based voltage-mode filters in [19,20].

Factor	[19]	[20]	Proposed
Number of single-ended output OTAs	6	6	5
Number of R + C	0 + 2	2 + 2	0 + 2
Resistorless structure	yes	no	yes
Marching condition (R = 1/g_m_)	no	yes	no
All grounded passive components	yes	yes	yes
Absent from using inverting-type input	yes	yes	yes
High impedance at the inputs	yes	yes	yes
Independent adjustment of ω_o_ and Q	no	yes	yes
Offer five standard filters	yes	yes	yes
Simultaneous realization of inverting/non-inverting lowpass, bandpass and bandreject filters	no	no	yes
Independent control of the inverting/non-inverting lowpass and bandpass gains simultaneously without affecting the parameters of ω_o_ and Q	no	no	yes
Easily transformed into a quadrature oscillator	no	no	yes
Supply voltages (V)	±5	±15	±2 ~ ±15
Power consumption (mW)	none	none	114 ~861
Output dynamic range (mV_p_)	±30	±30	±30 ~ ±100
Output power at 1dB compression, P1dB (dBm)	none	none	−14.6 ~ −9.2

**Table 2 sensors-19-02349-t002:** Single-input three-output filtering functions realized.

Input Conditions	Output Terminals	Filter Functions	Controllable Gain	Controllable Gain without Effecting ω_o_ and Q
Only V_i1_ as input	V_o1_	IBP	yes	no
	V_o2_	LP	no	no
	V_o3_	BR	no	no
Only V_i2_ as input	V_o1_	IBP	yes	yes
	V_o2_	LP	yes	yes
	V_o3_	IBP	yes	yes
Only V_i3_ as input	V_o1_	BP	yes	yes
	V_o2_	ILP	yes	yes
	V_o3_	BP	yes	yes
Only V_i5_ as input	V_o1_	BP	no	no
	V_o2_	ILP	yes	no
	V_o3_	IBR	yes	no

**Table 3 sensors-19-02349-t003:** Fixed output V_o3_ and various functions realized.

Input Conditions					Fixed Output V_o3_ Filtering Functions
V_i1_	V_i2_	V_i3_	V_i4_	V_i5_	
1	0	0	0	0	BR
0	1	0	0	0	IBP
0	0	1	0	0	BP
0	0	0	1	0	LP
0	0	0	0	1	IBR
0	0	0	1	1	IHP
1	1	0	0	0	AP

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
