# Peer review of "Versatile Tunable Voltage-Mode Biquadratic Filter and Its Application in Quadrature Oscillator"

_sensors, 2019, doi:10.3390/s19102349_

Round 1

Reviewer 1 Report

Well written paper. I would like to see a demonstration of the electronic tunability.

Reviewer 2 Report

The paper presents a tunable biquadratic filter and its applications in quadrature oscillator. According to the reviewer’s opinion, the paper is well-structured and clear. The topic is interesting and falls within the aim of the journal. In addition, the results are well-presented and could be helpful to further develop the same topic. Therefore, the paper can be accepted for publication in the current form.

Reviewer 3 Report

The paper is well written and includes simulated and experimental results. However, I have the following concerns and comments:

-          For which specific sensor applications we need such a universal filter that use high voltage supply and consume high power?.  

-          I think that the authors should present a specific sensor application for their circuit.

-          The simulated THD and the 3rd IMD are very high for very limited input signal (see figs 9 and 10). 4% THD for 120mV and 3.5% 3rd IMD  for 70mV while the voltage supply is +-15V.

-          The measured 3rd IMD (Fig. 18) show even worst value only -34.54dB for 30mVp.

-          What is the value of dynamic range of the filter? I think it will be quite limited. This should be added and compared to other designs.

-          What is the value of the power consumption of the filter? This should be added and compared to other designs.

Round 2

Reviewer 3 Report

The paper can be accepted now.